# Localized random projections challenge benchmarks for bio-plausible deep learning

## Abstract

Similar to models of brain-like computation, artificial deep neural networks rely on distributed coding, parallel processing and plastic synaptic weights. Training deep neural networks with the error-backpropagation algorithm, however, is considered bio-implausible. An appealing alternative to training deep neural networks is to use one or a few hidden layers with fixed random weights or trained with an unsupervised, local learning rule and train a single readout layer with a supervised, local learning rule. We find that a network of leaky-integrate-and-fire neurons with fixed random, localized receptive fields in the hidden layer and spike timing dependent plasticity to train the readout layer achieves 98.1% test accuracy on MNIST, which is close to the optimal result achievable with error-backpropagation in non-convolutional networks of rate neurons with one hidden layer. To support the design choices of the spiking network, we systematically compare the classification performance of rate networks with a single hidden layer, where the weights of this layer are either random and fixed, trained with unsupervised Principal Component Analysis or Sparse Coding, or trained with the backpropagation algorithm. This comparison revealed, first, that unsupervised learning does not lead to better performance than fixed random projections for large hidden layers on digit classification (MNIST) and object recognition (CIFAR10); second, networks with random projections and localized receptive fields perform significantly better than networks with all-to-all connectivity and almost reach the performance of networks trained with the backpropagation algorithm. The performance of these simple random projection networks is comparable to most current models of bio-plausible deep learning and thus provides an interesting benchmark for future approaches.

## 1 Introduction

While learning a new task, synapses deep in the brain undergo task-relevant changes (Hayashi-Takagi et al., 2015). These synapses are often many neurons downstream of sensors and many neurons upstream of actuators. Since the rules that govern such changes deep in the brain are poorly understood, it is appealing to draw inspiration from deep artificial neural networks (DNNs) (LeCun et al., 2015). DNNs and the cerebral cortex process information in multiple layers of many neurons (Yamins & DiCarlo, 2016; Kriegeskorte, 2015) and in both, the artificial and the biological neural networks, learning depends on changes of synaptic strengths (*Hebbian theory*, Hebb (1949)). However, learning rules in the brain are most likely different from the backpropagation algorithm (Crick, 1989; Marblestone et al., 2016; Rumelhart et al., 1986). Furthermore, biological neurons communicate by sending discrete spikes as opposed to real-valued numbers used in DNNs. Differences like these suggest that there exist other, possibly equally powerful, algorithms that are capable to solve the same tasks by using different, more biologically plausible mechanisms. Thus, an important question in computational neuroscience is how to explain the fascinating learning capabilities of the brain with bio-plausible network architectures and learning rules. On the other hand, from a pure machine learning perspective there is increasing interest in neuron-like architectures with local learning rules, mainly motivated by the current advances in neuromorphic hardware (Nawrocki et al., 2016).

Image recognition is a popular task to test the proposed models. Because of its relative simplicity and popularity, the MNIST dataset (28×28-pixel grey level images of handwritten digits, LeCun

(1998)) is often used for benchmarking. Typical performances of existing models are around 97-99% classification accuracy on the MNIST test set (see section 2 and Table 8). This value lies in the region of the benchmarks for a large class of classical DNNs trained with backpropagation but without data-augmentation or convolutional layers (see table in LeCun (1998)). Thus, accuracies around this value are assumed to be an empirical signature of backpropagation-like deep learning (Lillicrap et al., 2016; Sacramento et al., 2017). It is noteworthy, however, that several of the most promising approaches that perform well on MNIST have been found to fail on harder tasks (Bartunov et al., 2018).

An alternative to supervised training of all layers with backpropagation are fixed random weights, as proposed by general approximation theory (Barron, 1993) and the extreme learning field (Huang et al., 2006), or unsupervised training in the first layers, combined with supervised training of a read-out layer. Unsupervised methods are appealing since they can be implemented with local learning rules, see e.g. "Oja's rule" (Oja, 1982; Sanger, 1989) for principal component analysis or algorithms in Olshausen & Field (1997); Rozell et al. (2008); Liu & Jia (2012); Brito & Gerstner (2016) for sparse coding. A single readout layer can also be implemented with a local delta-rule (also called "perceptron rule"), which may be implemented by pyramidal spiking neurons with dendritic prediction of somatic spiking (Urbanczik & Senn, 2014). Since it is pointless to simply stack multiple fully connected layers trained with principal component analysis or sparse coding (Olshausen & Field, 1997) we investigate here networks with a single hidden layer.

The main objective of this study was to see how far we can go with a single hidden layer and local learning rules in networks of spiking neurons. To support the design choices of the spiking model, we compared the classification performance of different rate networks: networks trained with back-propagation, networks where the hidden layer is trained with unsupervised methods, and networks with fixed random projections in the hidden layer. Since sparse connectivity is sometimes superior to dense connectivity (Litwin-Kumar et al., 2017; Bartunov et al., 2018) and successful convolutional networks leverage local receptive fields, we investigated also sparse connectivity between input and hidden layer, where each hidden neuron receives input only from a few neighboring pixels of the input image.

## 2 RELATED WORK

In recent years, many bio-plausible approaches to deep learning have been proposed (see e.g. Marblestone et al. (2016) for a review). For achieving performances similar to deep learning methods, existing approaches usually use either involved architectures or elaborate mechanisms to approximate the backpropagation algorithm. Examples include the use of convolutional layers (Tavanaei & Maida (2016); Lee et al. (2018); Kheradpisheh et al. (2018) and table therein), dendritic computations (Hussain et al., 2014; Guergiuev et al., 2016; Sacramento et al., 2017) or approximations of the backpropagation algorithm such as feedback alignment (Lillicrap et al., 2016; Baldi et al., 2016; Nøkland, 2016; Samadi et al., 2017; Kohan et al., 2018; Bartunov et al., 2018) equilibrium propagation (Scellier & Bengio, 2017), membrane potential based backpropagation (Lee et al., 2016), restricted Boltzmann machines and deep belief networks (O'Connor et al., 2013; Neftci et al., 2014), (localized) difference target propagation (Lee et al., 2015; Bartunov et al., 2018), reinforcement-signal models like AuGMEnT (Rombouts et al., 2015) or approaches using predictive coding (Whittington & Bogacz, 2017). Many models implement spiking neurons to stress bio-plausibility (Liu et al. (2016); Neftci et al. (2017); Kulkarni & Rajendran (2018); Wu et al. (2018); Liu & Yue (2018) and table therein) or for coding efficiency (O'Connor et al., 2017). The conversion of DNNs to spiking neural networks (SNN) after training with backpropagation (Diehl et al., 2015) is a common technique to evade the difficulties of training with spikes. Furthermore, there are models including recurrent activity (Spoerer et al., 2017; Bellec et al., 2018) or even starting directly from realistic circuits (Delahunt & Kutz, 2018). We refer to Table 8 for a list of current bio-plausible MNIST benchmark models.

## 3 RESULTS

We study networks that consist of an input ($l_0$), one hidden ($l_1$) and an output-layer ($l_2$) connected by weight matrices $\mathbf{W}_1$ and $\mathbf{W}_2$ (Figure 1). Training the hidden layer weights $\mathbf{W}_1$ with standard

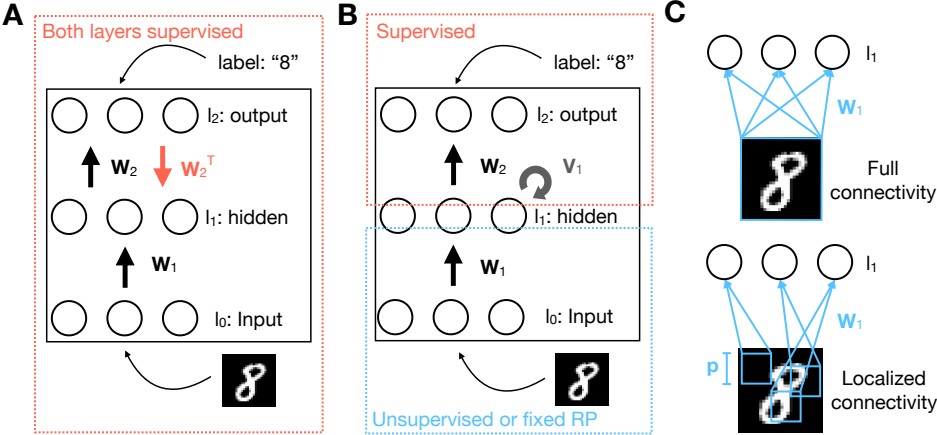

Figure 1: The network model. **A** Training with Backpropagation (BP) through one hidden layer. **B** Architecture with unsupervised feature learning or fixed Random Projections (RP) in the first layer and a supervised classifier in the second layer. **W** stands for feed-forward, **V** for recurrent, inhibitory weights (Only used for unsupervised feature learning of the first layer weights $\mathbf{W}_1$). **C** Illustration of fully connected and localized receptive fields of $\mathbf{W}_1$.

supervised training involves (non-local) error backpropagation using the transposed weight matrix $\mathbf{W}_2^T$ (Figure 1A). In the bio-plausible network considered in this paper (Figure 1B), the input-to-hidden weights $\mathbf{W}_1$ are either learned with an unsupervised method (Principal Component Analysis or Sparse Coding) or are fixed random projections. The unsupervised methods assume recurrent inhibitory weights $\mathbf{V}_1$ between hidden units to implement competition.

### 3.1 SPIKING LOCALIZED RANDOM PROJECTIONS

We first present the results with networks of leaky integrate-and-fire (**LIF**) neurons. The network architecture is as in Figure 1B, but without the recurrent connections $\mathbf{V}_1$. For implementing localized Random Projections (*l*-**RP**) in the hidden layer weights $\mathbf{W}_1$, we first chose the centers of the localized receptive fields at random positions in the input space and then randomly chose the weights therein, see Figure 1C. The receptive field patches span $p \times p$ pixels around their center position (we used $p = 10$ for the 28×28-pixel MNIST data). The output layer weights $\mathbf{W}_2$ are trained with a supervised spike timing dependent plasticity (**STDP**) rule.

#### 3.1.1 LIF AND STDP DYNAMICS

The spiking dynamics follow the usual LIF equations (see methods A.4) and the readout weights $\mathbf{W}_2$ evolve according to a supervised STDP delta rule using post-synaptic spike-traces $\mathrm{tr}_i(t)$ and a post-synaptic target trace $\mathrm{tgt}_i(t)$

$$\tau_{\mathrm{tr}} \frac{d\mathrm{tr}_i(t)}{dt} = -\mathrm{tr}_i(t) + \sum_f \delta\left(t - t_i^f\right)$$

$$\Delta w_{2,ij} = \alpha \cdot \left(\mathrm{tgt}_i^{\mathrm{post}}(t) - \mathrm{tr}_i^{\mathrm{post}}(t)\right) \delta\left(t - t_j^f\right). \tag{1}$$

Thus, for a specific readout weight $w_{2,ij}$, the post-synaptic trace is updated at every post-synaptic spike time $t_i^f$ and the weight is updated at every pre-synaptic spike time $t_j^f$. The target trace is used for feeding in the one-hot coded, supervisory signal for the MNIST classification into the output layer ($l_2$).

For a proof-of-principle and efficient parameter search we first investigate an LIF rate model. This rate model mimics the LIF dynamics by using the LIF activation function $\varphi_{\mathrm{LIF}}$ as nonlinearity,

$$\mathrm{rate}(u) = \varphi_{\mathrm{LIF}}(u) = \left[\Delta_{\mathrm{abs}} - \tau_m \ln\left(1 - \frac{\vartheta}{u}\right)\right]^{-1}, \tag{2}$$

where $u$ is the membrane potential, $\Delta_{\text{abs}}$ the refractory period, $\tau_m$ the membrane time constant and $\vartheta$ the firing threshold of the LIF model. Furthermore, it employs the rate-version of the STDP delta rule Equation 1 (see methods section A.4 for details)

$$\Delta w_{ij} = \tilde{\alpha} \cdot \text{rate}_j^{\text{pre}} \cdot \left( \text{tgtrate}_i^{\text{post}} - \text{rate}_i^{\text{post}} \right), \tag{3}$$

where $\text{tgtrate}_i^{\text{post}}$ is the post-synaptic target rate, corresponding to the post-synaptic target trace $\text{tgt}_i(t)$ in Equation 1. We obtained similar spiking and weight dynamics when the readout weights $\mathbf{W}_2$ were either directly trained with STDP or trained with the LIF rate model and then plugged into the spiking LIF network (as done in e.g. Diehl et al. (2015)).

To illustrate the LIF and STDP dynamics, a toy example consisting of one pre- connected to one post-synaptic neuron was integrated for 650 ms. The pre- and post-synaptic membrane potentials show periodic spiking (Figure 2A) which induces post-synaptic spike traces and corresponding weight changes (Figure 2B), according to Equation 1. For the MNIST task, Figure 2C shows a raster plot for an exemplary training and testing protocol. During activity transients after pattern switches, learning is disabled until regular spiking is recovered. This is done, first, to ensure stability during activity transients (see Naud et al. (2008) and references therein) and second, to achieve decorrelation between the activities of subsequent patterns, as needed for stochastic gradient descent (SGD). During the testing period, learning is shut off permanently (see methods section A.4 for more details).

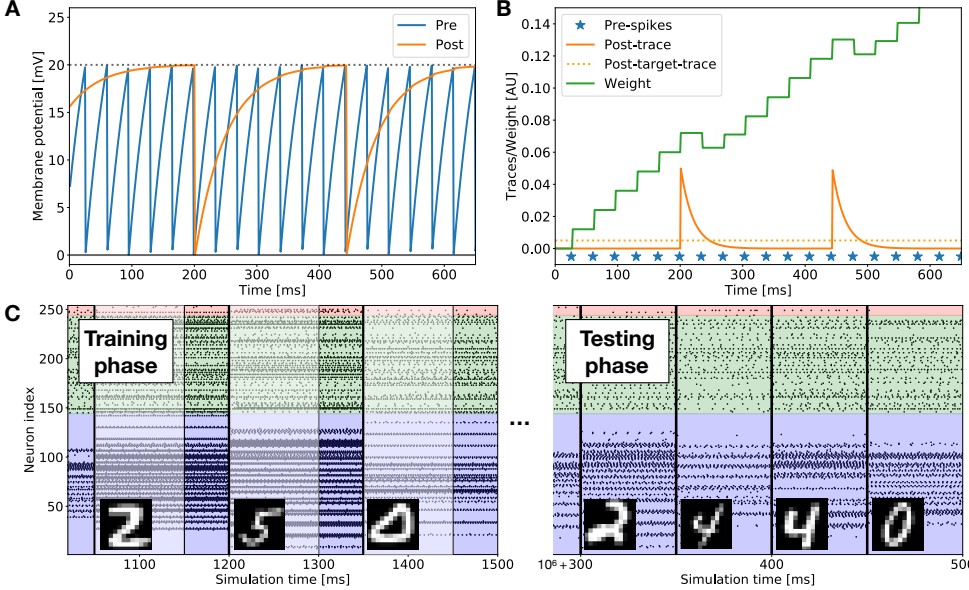

Figure 2: Spiking LIF and STDP dynamics. **A** Dynamics of the pre- and postsynaptic membrane potentials, spike-traces and the weight value (**B**) of a toy example with two neurons and one synapse. The weight decreases when the post-trace is above the post-target-trace (c.g. Equation 1 and subsection A.4). Both neurons receive static external input: $I_{\text{pre}}^{\text{ext}} \gg I_{\text{post}}^{\text{ext}} \approx \vartheta$ (spiking threshold). **C** Rasterplot of a network trained on MNIST, where every spike is marked with a dot. The background color indicates the corresponding layers: input (blue, $n_0 = 100$ neurons), hidden (green, $n_h = 100$) and output (red, $n_2 = 10$). Bold vertical lines indicate pattern switches, thin lines indicate ends of transient phases (indicated by semi-transparency), during which learning is disabled. Left: Behaviour at the beginning of the training phase. Right: Testing period (learning off) after $10^4$ iterations (presented patterns), which is 1/6 of an epoch. The output layer has started to learn useful, 1-hot encoded class predictions. A downsampled ($12 \times 12$) version of MNIST was used for improved visibility. Other parameters, see appendix Appendix B.

### 3.1.2 CLASSIFICATION RESULTS FOR LIF $l$-RP

When directly trained with the STDP rule in Equation 1 the spiking LIF $l$-RP model ($n_h = 5000$ hidden units and patch size $p = 10$) reaches 98.1% test accuracy on MNIST. The corresponding LIF

rate model reaches 98.5% test accuracy. Transferring weights learned with the LIF rate model into the spiking LIF model resulted in similar accuracies as the LIF rate model. Table 1 compares the performances of the rate and spiking LIF $l$-RP models with the reference algorithm $l$-BP, which is a rate model trained with backpropagation, see subsection 3.2 and subsection 3.3 (for same hidden layer size $n_h$ and patch size $p$). We can see that the spiking LIF model almost reaches the performance of the corresponding rate model. The remaining gap (0.4%) between rate and spiking LIF model presumably stems from transients and the shorter training time of the spiking model (only $10^6$ compared to $10^7$ iterations due to long simulation times). Both, the rate and spiking LIF model of $l$-RP achieve accuracies close to the backpropagation reference algorithm $l$-BP and certainly lie in the range of current bio-plausible MNIST benchmarks, i.e. 97-99% test accuracy (see section 2 and Table 8). Based on these numbers we conclude that the spiking LIF model of localized random projections using STDP is capable of learning the MNIST task to a level that is competitive with known benchmarks for spiking networks.

| $l$-RP spiking LIF | $l$-RP rate LIF | $l$-BP |
|:---:|:---:|:---:|
| $98.1 \pm 0.04$ | $98.5 \pm 0.16$ | $98.8 \pm 0.1$ |

Table 1: Test accuracies (%) on MNIST for $n_h$ = 5000 hidden neurons and receptive field size $p$ = 10. The reference algorithm $l$-BP is a rate model trained with backpropagation, see subsection 3.2 and subsection 3.3. The rate models were trained with $10^7$ iterations (pattern presentations), the spiking LIF model with $10^6$ iterations.

## 3.2 BENCHMARKING RATE MODELS TRAINED WITH UNSUPERVISED LEARNING AND BACKPROPAGATION

To justify the design choices of the spiking model, we systematically investigated rate models with different methods to initialize or learn the hidden layer weights $\mathbf{W}_1$ (see Figure 1 and methods subsection A.1 for details). To set these hidden layer weights, we use either one of the unsupervised methods Principal Component Analysis (**PCA**) or Sparse Coding (**SC**), or train only the readout layer $\mathbf{W}_2$ and use fixed Random Projections (**RP**, as in subsection 3.1) for the hidden layer weights $\mathbf{W}_1$ (see Figure 1B). All these methods can be implemented with local, bio-plausible learning rules (Oja, 1982; Olshausen & Field, 1997). As a reference and upper performance bound, we train networks with the same architecture with standard backpropagation (**BP**, see Figure 1A). As a more bio-plausible approximation of BP, we include Feedback Alignment (**FA**, Lillicrap et al. (2016)) which uses fixed random feedback weights for error-backpropagation (see methods subsection A.3 for further explanation). A Simple Perceptron (**SP**) without a hidden layer serves as a simplistic reference, since it corresponds to direct classification of the input.

The hidden-to-output weights $\mathbf{W}_2$ are trained with standard stochastic gradient descent (SGD), using a one-hot representation of the class label as target. Since no error-backpropagation is needed for a single layer, the learning rule is local ("delta" or "perceptron"-rule, similar to Equation 3 of the LIF rate model). Therefore the system as a whole is bio-plausible in terms of online learning and synaptic updates using only local variables. For computational efficiency, we train first the hidden layer and then the output layer, however, both layers could be trained simultaneously.

We compared the test errors on the MNIST digit recognition data set for varying numbers of hidden neurons $n_h$ (Figure 3). The green PCA curve in Figure 3 ends at the vertical line $n_h = d = 784$ because the number of principal components (PCs), i.e. the number of hidden units $n_h$, is limited by the input dimension $d$. Since the PCs span the subspace of highest variance, classification performance quickly improves when adding more PCs for small $n_h$ and then saturates for larger $n_h$, crossing the (dotted) Simple Perceptron line at $n_h = 25$ PC hidden neurons. This intersection and other measures of effective dimensionality (see methods subsection A.1) suggest that the MNIST dataset lies mostly in a low-dimensional linear subspace with $d_{\text{eff}} \approx 25 \ll d$.

SC performance (red curve) starts at a higher test error but improves as quickly with $n_h$ as PCA. With overcomplete representations ($n_h > d$), the network achieves a remarkable classification performance of around 96 % test accuracy. This suggests that the sparse representation and the features extracted by SC are indeed useful for classification, especially in the overcomplete case.

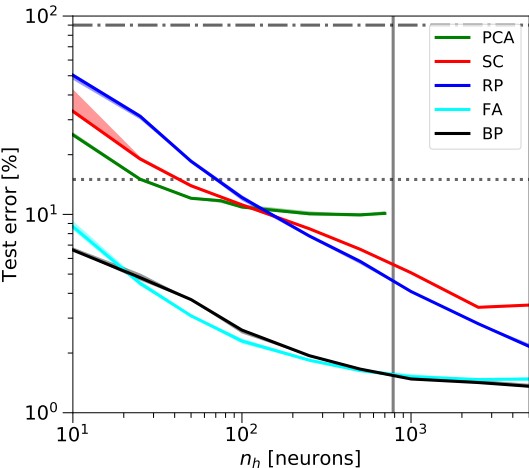

Figure 3: MNIST classification with rate networks, according to Figure 1. The test error decreases for increasing hidden layer size $n_h$ for Principal Component Analysis (PCA), Sparse Coding (SC), fixed Random Projections (RP) and the fully supervised reference algorithms Backpropagation (BP) and Feedback Alignment (FA). The dashed dotted line at 90 % is chance level, the dotted line around 14 % is the performance of a Simple Perceptron without hidden layer. The vertical line marks the input dimension $d = 784$, i.e. the transition from under- to overcomplete hidden representations. Note the log-log scale.

The performance of RP (blue curve) for small numbers of hidden units ($n_h < d$) is worse than for feature extractors like PCA and SC. Also for large hidden layers, performance improves only slowly with $n_h$, which is in line with theory (Barron, 1993) and findings in the extreme learning field (Huang et al., 2006). However, for large hidden layers sizes, RP outperforms SC. This suggests that the high dimensionality of the hidden layers is more important for reaching high performance than the features extracted by PCA or SC. Tests on the object recognition task CIFAR10 lead to the same conclusion, indicating that this observation is not entirely task specific (see subsection 3.3 for further analysis on CIFAR10). For all tested methods and hidden layer sizes, performance is significantly worse than the one reached with BP (black curve in Figure 3). In line with (Lillicrap et al., 2016), we find that FA (cyan curve) performs as well as BP on MNIST.

Universal function approximation theory predicts lower bounds for the squared error that follow a power law with hidden layer size $n_h$ for both BP ($\mathcal{O}(1/n_h)$) and RP ($\mathcal{O}(1/n_h^{2/d})$), where $d$ is the input dimension Barron et al. (1994); Barron (1993)). In the log-log-plot in Figure 3 this would correspond to a factor $d/2 = 784/2 = 392$ between the slopes of the curves of BP and RP, or at least a factor $d_{\text{eff}}/2 \approx 10$ using an effective dimensionality of MNIST (see methods A.1). We find a much faster decay of classification error in RP and a smaller difference between RP and BP slopes than suggested by the theoretical lower bounds.

### 3.3 LOCALIZED RANDOM RECEPTIVE FIELDS

There are good reasons to reduce the connectivity from all-to-all to localized receptive fields (Figure 1C): local connectivity patterns are observed in real neural circuits (Hubel & Wiesel, 1962), proven useful theoretically (Litwin-Kumar et al., 2017) and empirically (Bartunov et al., 2018), and successfully used in convolutional networks (CNNs). Even though this modification seems well justified from both biological and algorithmic sides, it reduces the generality of the algorithm to input data such as images where neighborhood relations between pixels (i.e. input dimensions) are important.

For random projections with localized receptive fields ($l$-**RP**), the centers of the patches were chosen at random positions in the input space and their weights where randomly fixed (as in subsection 3.1, see Figure 1C). We tested different patch sizes of $p \times p$ pixels and found an optimum around $p \approx 10$ which is more pronounced for large hidden layer sizes $n_h$ (see Figure 4A). Note that $p = 1$ corresponds to resampling the data with random weights, and $p = 28$ recovers fully connected RP performance.

The main finding here is the significant improvement in performance using $l$-RP: the optimum around $p \approx 10$ almost reaches BP performance for $n_h = 5000$ hidden neurons (blue arrow in

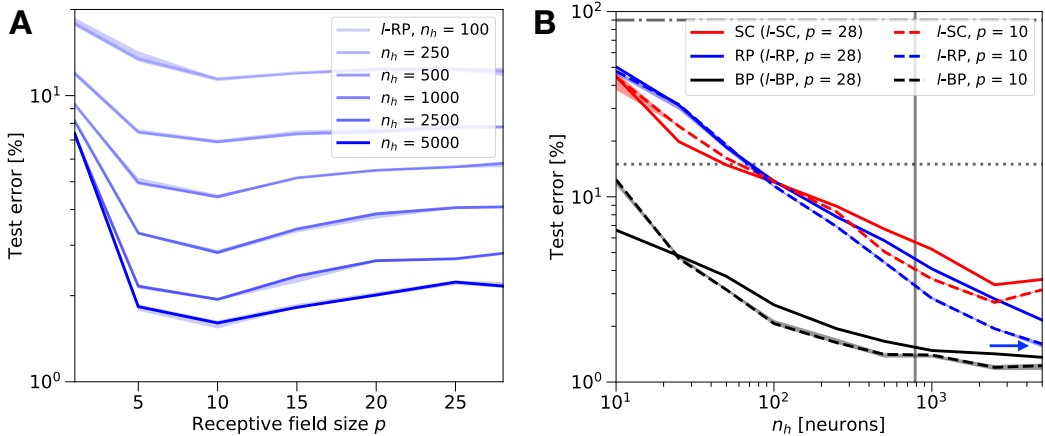

Figure 4: Effect of localized connectivity on MNIST. **A** Test error for localized Random Projections ($l$-RP), dependent on receptive field size $p$ for different hidden layer sizes $n_h$. The optimum at $p = 10$ is more pronounced for large hidden layer sizes. Full connectivity is equivalent to $p = 28$. Note the log-lin scale. **B** Localized receptive fields decrease test errors of Sparse Coding (SC), Random Projections (RP) and Backpropagation (BP). The effect is most significant for $l$-RP, which outperforms $l$-SC and almost approaches ($l$-)BP performance for large $n_h$ and $p = 10$ (blue arrow). All other reference lines as in Figure 3. Note the log-log scale.

Figure 4B). As expected $l$-RP and the LIF rate model of $l$-RP in subsection 3.1 perform equally well. To achieve a fair comparison BP and SC were also tested with localized receptive fields (*l-BP, l-SC*, see Figure 4B). Also these algorithms seem to benefit from localized connectivity (also with an optimum for patch size $p = 10$), however, not as much as RP. This makes $l$-RP a strong competitor of SC (and also FA, see Figure 3) as a bio-plausible algorithm in the regime of large, overcomplete hidden layers $n_h > d$.

Since classification performances of $l$-RP and $l$-BP are very close for layer sizes above $n_h = 5000$, we investigated the misclassified MNIST digits for both algorithms. We find that 75% of the ($\approx$ 125) misclassified digits of $l$-BP ($n_h = 5000$) are contained in the misclassified ones of $l$-RP ($n_h = 5000$). This means that in roughly 75% of the cases $l$-RP fails, also the reference algorithm $l$-BP fails, suggesting that these digits are particularly hard to recognize for networks with one hidden layer. We trained networks with up to $n_h = 100000$ hidden neurons to test if ($l$-)RP can finally reach ($l$-)BP performance, since the latter saturates for large $n_h$ (see Figure 4B). Indeed for simulations with $n_h = 100000$ and $p = 10$, $l$-BP and $l$-RP performance was not significantly different any more, both being at 1.2% test error.

To test whether $l$-RP only works for the relatively simple MNIST data set (centered digits, non-informative margin pixels, no clutter, uniform features and perspective etc.) or generalizes to more difficult tasks, we applied it to the CIFAR10 data set (Krizhevsky, 2013). We first reproduced a typical benchmark performance of a fully connected network with one hidden layer trained with standard BP ($\approx$ 56% test accuracy, $n_h = 5000$, see also Lin & Memisevic (2016)). Again, $l$-RP outperforms the unsupervised methods PCA and $l$-SC in the case of large, overcomplete hidden layers (see Table 2). Furthermore, as on MNIST, classification performance increases for increasing hidden layer size $n_h$ and localized receptive fields perform better than full connectivity for all methods.

Also on CIFAR10, $l$-RP comes close to the performance of the reference algorithm $l$-BP, however, the difference between $l$-RP and $l$-BP is larger than on MNIST. Given that state-of-the-art performance on the CIFAR10 dataset with deep convolutional neural networks is close to 98% (e.g. Real et al. (2018)), the limitations of $l$-RP and the difference in difficulty between MNIST and CIFAR10 become apparent.

## 4 DISCUSSION

The rules that govern plasticity of synapses deep in the brain remain elusive. In contrast to bio-plausible deep learning based on approximations of the backpropagation algorithm, we focused

|         | PCA            | $l$-SC         | $l$-RP        | $l$-BP        |
|---------|----------------|----------------|---------------|---------------|
| MNIST   | $90.3 \pm 0.02$ | $97.3 \pm 0.03$ | $98.4 \pm 0.1$ | $98.8 \pm 0.1$ |
| CIFAR10 | $22.6 \pm 0.003$ | $35.3 \pm 0.004$ | $52.0 \pm 0.004$ | $58.3 \pm 0.002$ |

Table 2: Test accuracies (%) on MNIST and CIFAR10. PCA uses full connectivity and 700 (2500) PCs for MNIST (CIFAR10). All other methods use $n_h = 5000$ hidden neurons and receptive field size $p = 10$. Note that CIFAR10 has $d = 32 \times 32 \times 3 = 3072$ input channels (the third factor is due to the color channels), MNIST only $d = 28 \times 28 = 784$. The models were trained for $10^7$ iterations ($\approx$ 167 epochs).

here on training a readout layer with a supervised, local learning rule combined with a single hidden layer with either fixed random weights or trained with unsupervised, local learning rules.

To our surprise, randomly initialized fixed weights (RP) of large hidden layers lead to better classification performance than training them with unsupervised methods like PCA or sparse coding (SC). This implies that the inductive bias of PCA and sparse coding is not well suited for the task of digit classification and object recognition. It may be interesting to search for alternative unsupervised, local learning rules with a stronger inductive bias.

Replacing all-to-all connectivity with localized input filters is such an inductive bias that was already seen to be useful in other models (Bartunov et al., 2018) and proved to be particularly useful in conjunction with randomly initialized static weights. Already for a hidden layer size of 5000 neurons the performance of $l$-RP almost reaches the performance of backpropagation on MNIST. Furthermore, performance scaling with the number of hidden units $n_h$ was found to be orders of magnitudes better than the lower bound suggested by universal function approximation theory (Barron, 1993).

Since we wanted to keep our models as simple as possible, we used online (no mini-batches) stochastic gradient descent (SGD) with a constant learning rate in all our experiments. There are many known ways to further tweak the final performance, e.g. with adaptive learning rate schedules or data augmentation, but our goal here was to demonstrate that even the simple model with localized random projections and spike timing dependent plasticity with a constant learning rate achieves results that are comparable with more elaborate approaches that use e.g. convolutional layers with weight sharing (Panda & Roy, 2016), backpropagation approximations (Lee et al., 2016), multiple hidden layers (Lillicrap et al., 2016), dendritic neurons (Sacramento et al., 2017), recurrence (Diehl & Cook, 2015) or conversion from rate to spikes (Diehl et al., 2015).

Above 98% accuracy we have to take into account a saturating effect of the network training: better models will only lead to subtle improvements in accuracy. It is not obvious whether improvements are really a proof of having achieved deep learning or just the result of tweaking the models towards the peculiarities of the MNIST dataset (centered digits, non-informative margin pixels, no clutter, uniform features and perspective etc.). We observed that more challenging data sets such as CIFAR10 clearly highlight the limitations of $l$-RP and thus are better suited to test deep learning capabilities. We are aware that state-of-the-art deep learning has moved from MNIST to harder datasets, such as ImageNet (Deng et al., 2009), long ago. Yet MNIST seems to be the current reference task for most bio-plausible deep learning models (see section 2 and Table 8).

In this paper we presented a new MNIST benchmark for bio-plausible spiking networks. Using localized random projections ($l$-RP) and STDP learning, our spiking LIF model reached 98.1% test accuracy on MNIST which lies within the range of current benchmarks for bio-plausible models for deep learning (see section 2 and Table 8). Our network model is particularly simple, i.e. it has only one trainable layer and does not depend on sophisticated architectural or algorithmic features (e.g. to approximate backpropagation). Instead it relies on the properties of high-dimensional localized random projections. We suggest that novel, progressive approaches to bio-plausible deep learning should significantly outperform the benchmark presented here.

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

# A    METHODS

## A.1    RATE NETWORK MODEL

We use a 3-layer (input $l_0$, hidden $l_1 = l_h$ and output $l_2$) feed-forward rate-based architecture with layer sizes ($n_0$ for input), $n_1$ (hidden) and $n_2$ (output, with $n_2 = \#$ classes). The layers are connected via weight matrices $\mathbf{W}_1 \in \mathbb{R}^{n_1 \times n_0}$ and $\mathbf{W}_2 \in \mathbb{R}^{n_2 \times n_1}$ and each neuron receives bias from the bias vectors $\mathbf{b}_1 \in \mathbb{R}^{n_1}$ and $\mathbf{b}_2 \in \mathbb{R}^{n_2}$ respectively (see Figure 1). The neurons themselves are nonlinear units with an element-wise, possibly layer-specific, nonlinearity $\mathbf{a}_i = \varphi_l(\mathbf{u}_i)$. The feed-forward pass of this model thus reads

$$
\begin{aligned}
\mathbf{u}_{l+1} &= \mathbf{W}_{l+1}\mathbf{u}_l + \mathbf{b}_{l+1} \\
\mathbf{a}_{l+1} &= \varphi_{l+1}(\mathbf{u}_{l+1}).
\end{aligned}
\tag{4}
$$

The simple perceptron (SP) only consists of one layer ($l_2$, $\mathbf{W}_2 \in \mathbb{R}^{n_2 \times n_0}$, $\mathbf{b}_2 \in \mathbb{R}^{n_2}$). The sparse coding (SC) model assumes recurrent inhibition within the hidden layer $l_1$. This inhibition is not modeled by an explicit inhibitory population, as required by Dale's principle (Dale, 1935), but direct, plastic, inhibitory synapses $\mathbf{V}_1 \in \mathbb{R}^{n_1 \times n_1}$ are assumed between neurons in $l_1$. Classification error variances in Figure 3 & Figure 4 are displayed as shaded, semi-transparent areas with the same colors as the corresponding curves. Their lower and upper bounds correspond to the 25% and 75% percentiles of at least 10 independent runs.

An effective dimensionality $d_{\text{eff}}$ of the MNIST data set can be obtained, e.g. via eigen-spectrum analysis, keeping 90% of the variance. We obtain values around $d_{\text{eff}} \approx 20$. The measure proposed in Litwin-Kumar et al. (2017) gives the same value $d_{\text{eff}} \approx 20$. Another measure is the crossing of the PCA curve with the Simple Perceptron line in Figure 3 at $n_h = 25 (= d_{\text{eff}})$. We checked that training a perceptron (1 hidden layer, $n_h = 1000$, $10^7$ iterations, ReLU, standard BP) on the first 25 PCs of MNIST leads to 1.7% test error (vs 1.5% test error on the full MNIST data). Together, these findings suggest that the MNIST dataset lies mostly in a low-dimensional linear subspace with $d_{\text{eff}} \approx 25 \ll d$. The MNIST (& CIFAR10) data was rescaled to values in [0,1] and mean centered, which means that the pixel-wise average over the data was subtracted from the pixel values of every image. The code for the implementation of our rate network model will be available online upon acceptance.

## A.2    UNSUPERVISED TECHNIQUES

### A.2.1    PRINCIPAL COMPONENT ANALYSIS (PCA)

In this paper we do not implement PCA learning explicitly as a neural learning algorithm but by a standard PCA algorithm (`https://github.com/JuliaStats/MultivariateStats.jl`). For $d$-dimensional data such algorithms output the values of the $n \leq d$ first principal components as well as the principal subspace projection matrix $\mathbf{P} \in \mathbb{R}^{n \times d}$. This matrix can directly be used as feedforward matrix $\mathbf{W}_1$ in our network since the lines of $\mathbf{P}$ correspond to the projections of the data onto the single principal components. In other words each neuron in the hidden layer $l_1$ extracts another principal component of the data.

Since PCA is a linear model, biases $\mathbf{b}_1$ were set to $\mathbf{0}$ and the nonlinearity was chosen linear, i.e. $\varphi_1(\mathbf{u}) = \mathbf{u}$. With this, we can write the (trained) feed-forward pass of the first layer of our PCA model as follows:

$$
\mathbf{a}_1 = \mathbf{u}_1 = \mathbf{W}_1 \cdot \mathbf{a}_0 \quad \text{with } \mathbf{W}_1 = \mathbf{P} \tag{5}
$$

Since the maximum number of PCs that can be extracted is the dimensionality of the data, $n_{\max} = d$, the number of neurons in the hidden layer $n_1$ is limited by $d$. This makes PCA unusable for overcomplete hidden representations as investigated for SC and RP.

Consistency between the used standard algorithm and neural implementations of PCA ("Sanger's" rule Sanger (1989)) was checked by comparing the extracted PCs and visualizing the learned projections (lines of $\mathbf{P}$) for the case of 30 extracted PCs, i.e. $n = 30$.

### A.2.2 SPARSE CODING (SC)

For $d$-dimensional data, SC aims at finding a dictionary $\mathbf{W} \in \mathbb{R}^{h \times d}$ of features that lead to an optimal representation $\mathbf{a}_1 \in \mathbb{R}^h$ which is sparse, i.e. has as few non-zero elements as possible. The corresponding optimization problem reads:

$$
\begin{aligned}
\mathbf{W}^{opt}, \mathbf{a}_1^{opt} &= \operatorname{argmin} \mathcal{L}(\mathbf{W}, \mathbf{a}_1) \\
\mathcal{L}(\mathbf{W}, \mathbf{a}_1) &= \frac{1}{2}\|\mathbf{a}_0 - \mathbf{W}^\top \mathbf{a}_1\|_2^2 + \lambda\|\mathbf{a}_1\|_1.
\end{aligned}
\tag{6}
$$

Since this is a nonlinear optimization problem with latent variables (hidden layer) it cannot be solved directly. Usually an iterative two step procedure is applied (akin to the expectation-maximization algorithm) until convergence: First optimize with respect to the activities $\mathbf{a}$ with fixed weights $\mathbf{W}$. Second, assuming fixed activities, perform a gradient step w.r.t to weights.

We implement a biologically plausible SC model using a 2-layer network with recurrent inhibition and local plasticity rules similar to the one in Brito & Gerstner (2016). For a rigorous motivation (and derivation) that such a network architecture can indeed implement sparse coding we refer to Olshausen & Field (1997); Zylberberg et al. (2011); Pehlevan & Chklovskii (2015); Brito & Gerstner (2016). We apply the above mentioned two step optimization procedure to solve the SC problem given our network model. The following two steps are repeated in alternation until convergence of the weights:

1. **Optimizing the hidden activations:**
   We assume given and fixed weights $\mathbf{W}_1$ and $\mathbf{V}_1$ and ask for optimal hidden activations $\mathbf{a}_1$. Because of the recurrent inhibition $\mathbf{V}_1$ the resulting equation for the hidden activities $\mathbf{a}_1$ is nonlinear and implicit. To solve this equation iteratively, we simulate the dynamics of a neural model with time-dependent internal and external variables $\mathbf{u}_1(t)$ and $\mathbf{a}_1(t)$ respectively. The dynamics of the system is then given by Zylberberg et al. (2011); Brito & Gerstner (2016):

$$
\begin{aligned}
\tau_u \frac{d\mathbf{u}_1(t)}{dt} &= -\mathbf{u}_1(t) + (\mathbf{W}_1 \mathbf{a}_0(t) - \mathbf{V}_1 \mathbf{a}_1(t)) \\
\mathbf{a}_1(t) &= \varphi(\mathbf{u}_1(t))
\end{aligned}
\tag{7}
$$

   In practice the dynamics is simulated for $N_{\text{iter}} = 50$ iterations, which leads to satisfying convergence (change in hidden activations $< 5\%$).

2. **Optimizing the weights:**
   Now the activities $\mathbf{a}_1$ are kept fixed and we want to update the weights following the gradient of the loss function. The weight update rules are Hebbian-type local learning rules (Brito & Gerstner, 2016):

$$
\begin{aligned}
\Delta W_{1,ji} &= \alpha_w \cdot a_{0,i} \cdot a_{1,j} \\
\Delta V_{1,jk} &= \alpha_v \cdot a_{1,k} \cdot (a_{1,j} - \langle a_{1,j} \rangle)
\end{aligned}
\tag{8}
$$

   $\langle \cdot \rangle$ is a moving average (low-pass filter) with some time constant $\tau_{\text{mav}}$. At the beginning of the simulation (or after a new pattern presentation) $\tau_{\text{mav}}$ is increased starting from 0 to $\tau_{\text{mav}}$ during the first $\tau_{\text{mav}}$. The values of the lines of $\mathbf{W}_1$ are normalized after each update, however this can also be achieved by adding a weight decay term. Additionally the values of $\mathbf{V}_1$ are clamped to positive values after each update to ensure that the recurrent input is inhibitory. Also the diagonal of $\mathbf{V}_1$ is kept at zero to avoid self-inhibition.

During SC learning, at every iteration, the variabes $\mathbf{u}_1(t)$ and $\mathbf{a}_1(t)$ are reset (to avoid transients) before an input is presented. Then for every of the $N$ iterations, equation 7 is iterated for $N_{\text{iter}}$ steps

and the weights are updated according to equation 8.

For comparison with localized RP ($l$-RP, see subsubsection A.2.3), a localized version of SC was implemented with the same initialization of $\mathbf{W}_1$ as in $l$-RP. The usual SC learning rule equation 8 is applied and the localized connectivity is kept by clamping weights outside the receptive fields to zero. Lateral inhibition weights $\mathbf{V}_1$ are initialized and learned as in normal SC (full competition is kept). For a detailed parameter list, see Table 3.

### A.2.3 RANDOM PROJECTIONS (RP)

For RP, the weight matrix $\mathbf{W}_1$ between input and hidden layer is initialized randomly $\mathbf{W}_1 \sim \mathcal{N}(0, \sigma^2)$ with variance-preserving scaling: $\sigma^2 \propto 1/n_0$. The biases $\mathbf{b}_1$ are initialized by sampling from a uniform distribution $\mathcal{U}([0, 0.1])$ between 0 and 0.1. In practice we used the specific initialization

$$
\begin{aligned}
\mathbf{W}_1 &\sim \mathcal{N}(0, \sigma^2) \ \sigma^2 = \frac{1}{100\,n_0} \\
\mathbf{b}_1 &\sim \mathcal{U}([0, 0.1])
\end{aligned}
\tag{9}
$$

for RP (keeping weights fixed), SC, SP and also BP & RF (both layers with $\mathbf{W}_2, \mathbf{b}_2$ and $n_1$ respectively). The initialization of the biases $\mathbf{b}$ was found to be uncritical in the range of [0,0.1]

For localized RP ($l$-RP), neurons in the hidden layer receive input only from a fraction of the input units called a receptive field. Receptive fields are chosen to form a compact patch over neighbouring pixels in the image space. For each hidden neuron a receptive field of size $p \times p$ ($p \in \mathbb{N}$) input neurons is created at a random position in the input space. The weight values for each receptive field (rf) and the biases are initialized as:

$$
\begin{aligned}
\mathbf{W}_{1,\text{rf}} &\sim \mathcal{N}(0, \sigma_{\text{rf}}^2) \ \ \sigma_{\text{rf}}^2 = \frac{c}{100\,p}
\end{aligned}
\tag{10}
$$

$$
\mathbf{b}_1 \sim \mathcal{U}([0, 0.1])
\tag{11}
$$

were the optimization factor $c = 3$ was found empirically through a grid-search optimization of classification performance. For exact parameter values, see Table 4.

### A.3 CLASSIFIER & SUPERVISED REFERENCE ALGORITHMS

The connections $\mathbf{W}_2$ from hidden to output layer are updated by a simple delta-rule which is equivalent to BP in a single-layer network and hence is bio-plausible. For having a reference for our bio-plausible models (Figure 1B), we compare it to networks with the same architecture (number of layers, neurons, connectivity) but trained in a fully supervised way with standard backpropagation (Figure 1A). The forward pass of the model reads:

$$
\begin{aligned}
\mathbf{u}_{l+1} &= \mathbf{W}_{l+1}\mathbf{u}_l + \mathbf{b}_{l+1}
\end{aligned}
\tag{12}
$$

$$
\mathbf{a}_{l+1} = \varphi_{l+1}(\mathbf{u}_{l+1})
\tag{13}
$$

The error $\tilde{\mathbf{e}}_L$ is calculated from the comparison of activations in the last layer $\mathbf{a}_L$ with the (one-hot encoded) target activations $\mathbf{tgt}$, with respect to the chosen loss function: mean squared error (MSE),

$$
\tilde{\mathbf{e}}_L = \mathbf{tgt} - \mathbf{a}_L
\tag{14}
$$

$$
\mathcal{L}_{\text{MSE}} = \frac{1}{2}\|\mathbf{tgt} - \mathbf{a}_L\|_2^2
\tag{15}
$$

or softmax/cross-entropy loss (CE),

$$
\mathbf{p} = \text{softmax}(\mathbf{a}_L)
\tag{16}
$$

$$
\tilde{\mathbf{e}}_L = \mathbf{tgt} - \mathbf{p}
\tag{17}
$$

$$
\mathcal{L}_{\text{CE}} = -\sum_{i=1}^{n_L} \text{tgt}_i \cdot \log(p_i)
\tag{18}
$$

Classification results (on the test set) for MSE- and CE-loss were found to be not significantly different. Rectified linear units (ReLU) were used as nonlinearity $\varphi(\mathbf{u}_l)$ for all layers (MSE-loss) or for the first layer only (CE-loss).

In BP the weight and bias update is obtained by stochastic gradient descent, i.e. $\Delta W_{l,ij} \propto \frac{\partial \mathcal{L}}{\partial W_{l,ij}}$. The full BP algorithm for deep networks reads (Rumelhart et al., 1986):

$$
\begin{aligned}
\mathbf{e}_L &= \varphi'_L(\mathbf{u}_L) \odot \tilde{\mathbf{e}}_L \\
\mathbf{e}_{l-1} &= \varphi'_{l-1}(\mathbf{u}_l) \odot \mathbf{W}_l^\top \mathbf{e}_l \\
\Delta \mathbf{W}_l &= \alpha \cdot \mathbf{e}_l \otimes \mathbf{a}_{l-1} \\
\Delta \mathbf{b}_l &= \alpha \cdot \mathbf{e}_l
\end{aligned}
\tag{19}
$$

where $\odot$ stands for element-wise multiplication, $\otimes$ is the outer (dyadic) product, $\varphi'_l(\cdot)$ is the derivative of the nonlinearity and $\alpha$ is the learning rate. FA (Lillicrap et al., 2016) uses a fixed random matrix $\mathbf{R}_l$ instead of the transpose of the weight matrix $\mathbf{W}_l^\top$ for the error backpropagation step in equation 19.

To allow for a fair comparison with $l$-RP, BP and FA were implemented with full connectivity and with localized receptive fields with the same initialization as in $l$-RP. During training with BP (or FA), the usual weight update equation 19 was applied to the weights in the receptive fields, keeping all other weights at zero. The exact parameter values can be found in Table 5.

### A.4 SPIKING IMPLEMENTATION

### A.4.1 LIF MODEL

The spiking simulations were performed with a custom-made event-based leaky integrate-and-fire (LIF) integrator written in the `Julia`-language. For large network sizes, the exact, event-based integration can be inefficient due to a large frequency of events. To alleviate dramatic slow-down, an Euler-forward integration was added to the framework. For sufficiently small time discretization (e.g. $\Delta t \leq 5 \cdot 10^{-2}$ ms for the parameters given in Table 6) the error of this approximate integration does not have negative consequences on the learning outcome. Consistent results were obtained using event-based and Euler-forward integration. The code of this framework will be available online upon acceptance.

The dynamics of the LIF network is given by:

$$
\begin{aligned}
\tau_m \frac{du_i(t)}{dt} &= -u_i(t) + RI_i(t) \\
\text{with } I_i(t) &= I_i^{ff}(t) + I_i^{ext}(t) = \sum_{j,f} w_{ij}\epsilon\left(t - t_j^f\right) + I_i^{ext}(t)
\end{aligned}
$$

and the spiking condition: $\qquad$ if $u_i(t) \geq \vartheta_i$: $u_i \to u_{\text{reset}}$ $\qquad$ (20)

where $u_i(t)$ is the membrane potential, $\tau_m$ the membrane time-constant, $R$ the membrane resistance, $w_{ij}$ are the synaptic weights, $\epsilon(t) = \delta(t)/\tau_m$ (with $\tau_m$ in seconds) is the post-synaptic potential evoked by a pre-synaptic spike arrival, $\vartheta_i$ is the spiking threshold and $u_{\text{reset}}$ the reset potential after a spike. The input is split into a feed-forward ($I^{ff}(t)$) and an external ($I^{ext}(t)$) contribution. Each neuron in the input layer $l_0$ ($n_0 = d$) receives only external input $I^{ext}$ proportional to one pixel value in the data. To avoid synchrony between the spikes of different neurons, the starting potentials and parameters (e.g. thresholds) for the different neurons are drawn from a (small) range around the respective mean values.

We implement STDP using post-synaptic spike-traces $\text{tr}_i(t)$ and a post-synaptic target-trace $\text{tgt}_i(t)$.

$$
\begin{aligned}
\tau_{\text{tr}} \frac{d\text{tr}_i(t)}{dt} &= -\text{tr}_i(t) + \sum_f \delta\left(t - t_i^f\right) \\
\Delta w_{ij} &= g\left(\text{tr}_i^{\text{post}}(t), \text{tgt}_i^{\text{post}}(t)\right) \delta\left(t - t_j^f\right)
\end{aligned}
\tag{21}
$$

with the plasticity function

$$g\left(\text{tr}_i^{\text{post}}(t), \text{tgt}_i(t)\right) = \alpha \cdot \left(\text{tgt}_i^{\text{post}}(t) - \text{tr}_i^{\text{post}}(t)\right). \tag{22}$$

To train the network, we present patterns to the input layer and a target-trace to the output layer. The MNIST input is scaled by the input amplitude $\text{amp}_{\text{inp}}$, the targets $\textbf{tgt}(t)$ of the output layer are the one-hot-coded classes, scaled by the target amplitude $\text{amp}_{\text{tgt}}$. Additionally, every neuron receives a static bias input $I_{\text{bias}}^{\text{ext}} \approx \vartheta$ to avoid silent units in the hidden layer. Every pattern is presented as fixed input for a time $T_{\text{pat}}$ and the LIF dynamics as well as the learning evolves according to equation 20 and equation 21 respectively. To ensure stability during transients (see Naud et al. (2008) and references therein), learning is disabled after pattern switches for a duration of about $T_{\text{trans}} = 4\tau_m$. With the parameters we used for the simulations (see Table 6), firing rates of single neurons in the whole network stayed below 1 kHz which was considered as a bio-plausible regime. For the toy example in Figure 2A& B we used static input and target with the parameters $\text{amp}_{\text{inp}} = 40$, $\text{amp}_{\text{tgt}} = 5$ (i.e. target trace = 0.005), $\vartheta_{\text{mean}} = 20$, $\sigma_\vartheta = 0$, $\tau_m = 50$, $\alpha = 1.2 \cdot 10^{-5}$. For the raster plot in Figure 2C we used $\text{amp}_{\text{inp}} = 300$, $\text{amp}_{\text{tgt}} = 300$, $\vartheta_{\text{mean}} = 20$, $\sigma_\vartheta = 0$, $\tau_m = 50$, $\alpha = 1.2 \cdot 10^{-5}$ $T_{\text{pat}} = 50$ ms, $T_{\text{trans}} = 100$ ms.

### A.4.2 LIF RATE MODEL

The LIF dynamics can be mapped to a rate model described by the following equations:

$$
\begin{aligned}
\mathbf{u}_l &= \mathbf{W}_l \mathbf{u}_{l-1} + R\mathbf{I}^{ext} \\
\mathbf{a}_l &= \varphi_{\text{LIF}}\left(\mathbf{u}_l\right) \\
\Delta w_{ij} &= \tilde{g}\left(a_j^{\text{pre}}, a_i^{\text{post}}, \text{tgt}_i^{\text{post}}\right)
\end{aligned} \tag{23}
$$

with the (element-wise) LIF-activation function $\varphi_{\text{LIF}}(\cdot)$ and the modified plasticity function $\tilde{g}(\cdot)$:

$$\varphi_{\text{LIF}}\left(u_k\right) = \left[\Delta_{\text{abs}} - \tau_m \ln\left(1 - \frac{\vartheta_k}{u_k}\right)\right]^{-1} \tag{24}$$

$$\tilde{g}\left(a_j^{\text{pre}}, a_i^{\text{post}}, \text{tgt}_i^{\text{post}}\right) = \tilde{\alpha} \cdot a_j^{\text{pre}} \cdot \left(\text{tgt}_i^{\text{post}} - a_i^{\text{post}}\right) \tag{25}$$

The latter can be obtained by integrating the STDP rule Equation 21 and taking the expectation. Most of the parameters of the spiking- and the LIF rate models can be mapped to each other directly (see Tabs. 6 & 7). The learningrate $\alpha$ must be adapted since the LIF weight change depends on the presentation time of a pattern $T_{\text{pat}}$. In the limit of long pattern presentation times ($T_{\text{pat}} \gg \tau_m, \tau_{\text{tr}}$), the transition from the learning rate of the LIF rate model ($\tilde{\alpha}$) to the one of the spiking LIF model ($\alpha$) is

$$\alpha = \frac{1000 \text{ ms}}{T_{\text{pat}} \text{ [ms]}} \cdot 1000 \cdot \tilde{\alpha}, \tag{26}$$

where the second factor comes from a unit change from Hz to kHz. It is also possible to train weight matrices computationally efficient in the LIF rate model and plug them into the spiking LIF model afterwards (as in e.g. Diehl et al. (2015)). The reasons for the remaining difference in performance presumably lie in transients and single-spike effects that cannot be captured by the rate model. Also, the spiking network was only trained with $10^6$ image presentations (compared to $10^7$ for the rate model) due to long simulation times.

## B PARAMETER TABLES

In the following tables we use scientific E-notation $\text{XeY} = X \cdot 10^Y$ for better readability. For all simulations, we scaled the learning rate proportional to $1/n_h$ for $n_h > 5000$ to ensure convergence.

| Parameter | Description | Value |
|---|---|---|
| $n_h = n_1$ | Number of hidden units | [10,25,50,100,250,500,1000,2500,**5000**] |
| $p$ | Rec. field sizes (edge length) in units | [1,5,**10**,15,20,25,28] |
| $\alpha_w$ | Learning rate for $\mathbf{W}_1$ | 1e-3 |
| $\alpha_v$ | Learning rate for $\mathbf{V}_1$ | 1e-2 |
| $\alpha_c$ | Learning rate of classifier | 1e-2 |
| $\lambda$ | Sparsity parameter | [1e-4,1e-3,**1e-2**,1e-1,1e-0] |
| $S$ | Resulting sparsity (fraction of 0-elements in $l_1$) | 90 - 99% (dependent on $n_h$) |
| $\tau_{\mathrm{mav}}$ | Time constant of the moving average | 1e-2 [1/iterations] |
| $\tau_u$ | Time constant of inner variable $\mathbf{u}_1(t)$ | 1e-1 [1/iterations] |
| $N_{\mathrm{iter}}$ | Number of iterations solving eq. equation 7 | 50 |
| $N$ | Number of iterations for SC | 1e5 |
| $N_c$ | Num. of iterations for classifier training | 1e7 ($\approx$ 167 epochs) |
| $N_{\mathrm{inits}}$ | Number of trials for classifier | 5 |
| $\mathbf{W}_l^{\mathrm{init}}$ | Feed-forward weight initialization | $W_{l,ij} \sim \mathcal{N}(0,1)/(10\sqrt{n_{l-1}})$ |
| $\mathbf{V}_1^{\mathrm{init}}$ | Reccurent weight initialization | **0** |
| $\mathbf{b}_1^{\mathrm{init}}$ | Bias initialization | **0** (and kept fixed) |
| $\varphi_1(\cdot)$ | nonlinearity of hidden SC units | ReLU $\max(0, \cdot - \lambda)$ |
| $\varphi_2(\cdot)$ | nonlinearity of classifier | ReLU |

Table 3: (Hyper-)Parameters for SC. Best performing parameters in bold.

| Parameter | Description | Value |
|---|---|---|
| $n_h = n_1$ | Number of hidden units | [10,25,50,100,250,500,1000,2500,**5000**] |
| $p$ | Rec. field sizes (edge length) in units | [1,5,**10**,15,20,25,28] |
| $\alpha_l$ | Learning rate | 5e-3 |
| $N$ | Number of iterations | 1e7 ($\approx$ 167 epochs) |
| $N_{\mathrm{inits}}$ | Number of trials | 5 |
| $\mathbf{W}_l^{\mathrm{init}}$ | Feed-forward weight initialization | $W_{l,ij} \sim \mathcal{N}(0,1)/(10\sqrt{n_{l-1}})$ |
| $\mathbf{b}_1^{\mathrm{init}}$ | Bias initialization | $b_{l,i} \sim \mathcal{U}\left([0,1]\right)/10$ |

Table 4: (Hyper-)Parameters for RP. Best performing parameters in bold.

| Parameter | Description | Value |
|---|---|---|
| $n_h = n_1$ | Number of hidden units | [10,25,50,100,250,500,1000,2500,**5000**] |
| $p$ | Rec. field sizes (edge length) in units | [1,5,**10**,15,20,25,28] |
| $\alpha_l$ | Learning rate (per layer) | [5e-3,5e-3] (BP,FA) or 1e-2 (SP) |
| $N$ | Number of iterations | 1e7 ($\approx$ 167 epochs) |
| $N_{\mathrm{inits}}$ | Number of trials | 5 |
| $\mathbf{W}_l^{\mathrm{init}}$ | Feed-forward weight initialization | $W_{l,ij} \sim \mathcal{N}(0,1)/(10\sqrt{n_{l-1}})$ |
| $\mathbf{b}_1^{\mathrm{init}}$ | Bias initialization | $b_{l,i} \sim \mathcal{U}\left([0,1]\right)/10$ |

Table 5: (Hyper-)Parameters for BP, FA and SH. Best performing parameters in bold.

| Parameter | Description | Value |
|---|---|---|
| $n_h = n_1$ | Number of hidden units | [10,25,50,100,250,500,1000,2500,**5000**] |
| $p$ | Rec. field sizes (edge length) in units | [1,**10**,28] |
| $\tau_m$ | Membrane time constant | 25 ms |
| $R$ | Membrane resistance | 1 $\Omega$ |
| $\Delta_{abs}$ | Absolute refractory period | 0 ms |
| $\vartheta_i$ | Spiking thresholds | $\vartheta_i \sim \mathcal{N}(\vartheta_{mean}, \sigma_\vartheta)$ |
| $\vartheta_{mean}$ | Mean spiking threshold | 20 mV |
| $\sigma_\vartheta$ | Variance of spiking thresholds | 1 mV |
| $\mathrm{amp}_{inp}$ | Input amplitude | 500 mA |
| $\mathrm{amp}_{tgt}$ | Target amplitude | 500 mA |
| $I_{bias}^{ext}$ | External bias input to all neurons | $\vartheta_{mean}$/R |
| $\tau_{tr}$ | Spike trace time constant | 20 ms |
| $u_{reset}$ | Reset potential | 0 mV |
| $\alpha$ | Learning rate | 2e-4 ($n_h = 5000$, 5e-4 for Euler forward) |
| $N$ | Number of iterations | 1e6 ($\approx 17$ epochs) |
| $\mathbf{W}_l^{init}$ | Feed-forward weight initialization | $W_{l,ij} \sim \mathcal{N}(0,1) \cdot 20/\sqrt{n_{l-1}}$ |
| $T_{pat}$ | Duration of pattern presentation | 50 ms (train, 200 ms during testing) |
| $T_{trans}$ | Duration of the transient without learning | 100 ms |
| $\Delta t$ | Time step for Euler integrator | $\leq$ 5e-2 ms |

Table 6: (Hyper-)Parameters for the spiking LIF $l$-RP model. Input and target amplitudes are implausibly high due to the arbitrary convention $R = 1$ $\Omega$. Best performing parameters in bold.

| Parameter | Description | Value |
|---|---|---|
| $n_h = n_1$ | Number of hidden units | [10,25,50,100,250,500,1000,2500,**5000**] |
| $p$ | Rec. field sizes (edge length) in units | [1,**10**,28] |
| $\tau_m$ | Membrane time constant | 25 ms |
| $R$ | Membrane resistance | 1 $\Omega$ |
| $\Delta_{abs}$ | Absolute refractory period | 0 ms |
| $\vartheta_i$ | Spiking thresholds | $\vartheta_i \sim \mathcal{N}(\vartheta_{mean}, \sigma_\vartheta)$ |
| $\vartheta_{mean}$ | Mean spiking threshold | 20 mV |
| $\sigma_\vartheta$ | Variance of spiking thresholds | 1 mV |
| $\mathrm{amp}_{inp}$ | Input amplitude | 500 mA |
| $\mathrm{amp}_{tgt}$ | Target amplitude | 500 mA |
| $I_{bias}^{ext}$ | External bias input to all neurons | $\vartheta_{mean}$/R |
| $u_{reset}$ | Reset potential | 0 mV |
| $\tilde{\alpha}$ | Learning rate | 1e-8 (for $n_h = 5000$) |
| $N$ | Number of iterations | 1e7 ($\approx 167$ epochs) |
| $\mathbf{W}_l^{init}$ | Feed-forward weight initialization | $W_{l,ij} \sim \mathcal{N}(0,1) \cdot 20/\sqrt{n_{l-1}}$ |

Table 7: (Hyper-)Parameters for the LIF rate $l$-RP model. Parameters are the same as in the spiking model; only $\tilde{\alpha}$ was converted from $\alpha$ according to Equation 26. $10^7$ iterations were used for training the rate model (compared to $10^6$ in the spiking model). Input and target amplitudes are implausibly high due to the arbitrary convention $R = 1$ $\Omega$. Best performing parameters in bold.

## C  BIO-PLAUSIBLE MNIST BENCHMARKS

| Model | Neural coding | Learning type | Learning rule | Comments | Test accuracy (%) |
|---|---|---|---|---|---|
| Conv. SNN (Wu et al., 2018) | Spikes | Supervised | BP-variant | 5 conv. layers, Spatio-Temporal BP | 99.3 |
| Conv. SNN (Diehl et al., 2015) | Rate | Supervised | BP | Conversion: rate → spike | 99.1 |
| Conv. Spiking AE (Panda & Roy, 2016) | Spikes | Un/Supervised | membr.-potential based BP | Stacked conv. AE, weight sharing BP + sym. weights used inside AE | 99.1 |
| *l*-**FA** (Bartunov et al., 2018) (& this paper) | Rate | Supervised | FA | FA with localized rec. fields | **98.7** |
| SNN (Lee et al., 2016) | Spikes | Supervised | membr. pot. based BP | BP approx. weights symmetry | 98.7 |
| (Stoch.) Diff. Target Prop. (Lee et al., 2015) | Rate | Supervised | Targ. Prop. | Layer-wise AE | 98.5 |
| **LIF rate *l*-RP** (this paper) | Rate | Supervised | rate STDP | Only output layer learned | **98.5** |
| *l*-**RP** (this paper) | Rate | Supervised | Delta-rule | Only output layer learned | **98.4** |
| Conv. SNN (Kheradpisheh et al., 2018) | Spikes | Unsupervised | STDP | 3 Conv. layers, external SVM | 98.4 |
| (O'Connor et al., 2017) | Pseudo-spike | Supervised | Pseudo-STDP | Sparse, discrete activities | 98.3 |
| (Nøkland, 2016) | Rate | Supervised | direct FA | Many hidden layers | 98.3 |
| Spiking FA (Lillicrap et al., 2016) | Spikes | Supervised | FA | 3 hidden layers | 98.2 |
| **spiking LIF *l*-RP** (this paper) | Spikes | Supervised | STDP | Only output layer learned | **98.1** |
| Forward propagation (FP) (Kohan et al., 2018) | Rate | Supervised | FP & direct FA | FP: BP approximation | 98.1 |
| Spiking FA (Neftci et al., 2017) | Spikes | Supervised | FA | Direct FA | 98 |
| Predictive coding (Whittington & Bogacz, 2017) | Rate | Supervised | Local Hebbian | BP approximation by predictive coding | 98 |
| Spiking CNN (Tavanaei & Maida, 2016) | Rate/ Spikes | Unsupervised | SC, STDP | semi-online SVM | 98 |
| Equilibrium Prop. (Scellier & Bengio, 2017) | Rate | Supervised | EquiProp | 1 - 3 hidden layers | 97 - 98 |
| Dendr. BP (Sacramento et al., 2017) | Spikes | Supervised | Bio-plausible BP | Dendr. computation for BP approx. | 97.5 |
| *l*-**SC** (this paper) | Rate | Un/Supervised | SC/delta-rule | SC for 1. layer delta-rule for 2. | **97.4** |
| Spiking FA (Samadi et al., 2017) | Spikes | Supervised | FA | 3 hidden layers | 97 |
| Sparse/Skip FA (Baldi et al., 2016) | Rate | Supervised | FA | Sparse- & Skip-FA Lim. prec. FA | 96 - 97 |
| Spiking CNN (Thiele et al., 2018) | Spikes | Unsupervised | STDP | Recurren Inhib. Purely unsuperv. | 96.6 |
| Spiking FA (Guergiuev et al., 2016) | Spikes | Supervised | Bio-plausible BP | Dendr. computation for BP approx. | 96.3 |
| 2 layer network (Diehl & Cook, 2015) | Spikes | Unsupervised | STDP | Recurrent Inhib. Purely unsuperv. | 95 |
| Spiking RBM/DBN (O'Connor et al., 2013) | Rate | Supervised | Contrastive Divergence | Conversion rate → spike | 94.1 |
| 2 layer network (Querlioz et al., 2013) | Spikes | Unsupervised | STDP | Memristive device | 93.5 |
| Spiking HMAX/CNN (Liu & Yue, 2018) | Spikes | Supervised | event-driven cont. STDP | mod. HMAX model for preprocess. | 93 |
| Spiking RBM/DBN (Neftci et al., 2014) | Rate | Supervised | Contrastive Divergence | Neural sampling | 92.6 |
| Spiking RBM/DBN (Neftci et al., 2014) | Spikes | Supervised | Contrastive Divergence | Neural sampling | 91.9 |
| Spiking CNN (Zhao et al., 2015) | Spike | Supervised | Tempotron rule | Dynamic Vision Sensor MNIST | 91.3 |
| Dendritic neurons (Hussain et al., 2014) | Rate | Supervised | Morphology learning | Nonlin. dendrites Neuromorphic appl. | 90.3 |
| **PCA + class.** (this paper) | Rate | Un/Supervised | PCA-algorithm delta-rule | PCs as features | **90.3** |
| **SP** (this paper) | Rate | Supervised | Delta-rule | Direct class. on MNIST data | **85** |

Table 8: MNIST benchmarks for bio-plausible models of deep learning compared with models in this paper (**bold**). Models are ranked by accuracy (rightmost column). Accuracy refers to the classification accuracy on the MNIST test set. Parts of this table are taken from (Diehl & Cook, 2015) and (Kheradpisheh et al., 2018). Models involving convolutional/pooling layers are marked in blue. Note that the simple models in the *l*-RP class (*l*-RP, LIF rate & spiking *l*-RP), marked in red, perform better than several more elaborate models. For conventional ANN/DNN/CNN MNIST benchmarks see Table at http://yann.lecun.com/exdb/mnist/).

