# OpenReview forum: "Localized random projections challenge benchmarks for bio-plausible deep learning"
_ICLR.cc/2019/Conference_

### Official Review · AnonReviewer2 · 2018-11-02
**Single-layer SNN training on different unsupervised preprocessing**

**Rating:** 3
**Confidence:** 5

**Review:**

This article compares different methods to train a two-layer spiking neural network (SNN) in a bio-plausible way on the MNIST dataset, showing that fixed localized random connections that form the hidden layer, in combination with a supervised local learning rule on the output layer can achieve close to state-of-the-art accuracy compared to other SNN architectures. The authors investigate three methods to train the first layer in an unsupervised way: principal component analysis (PCA) on the rates, sparse coding of activations, and fixed random local receptive fields.  Each of the methods is evaluated on the one hand in a time-stepped simulator, using LIF neurons and on the other hand using a rate-approximated model which allows for faster simulations. Results are compared between each other and as reference with  standard backpropagation and feedback alignment.  The main finding is that localized random projections outperform other unsupervised ways of computing first layer features, and with many hidden neurons approaches backpropagation results. These results are summarized in Table 8, which compares results of the paper and other state-of-the-art and bio-plausible SNNs. PCA and sparse coding work worse on MNIST than local random projections, regardless if the network is rate-based, spike-based or a regular ANN trained with the delta rule. Feedback Alignment, although only meant for comparison, performs best of the algorithms investigated in this paper.

In general the question how to train multi-layer spiking neural networks in a bio-plausible way is very relevant for computational neuroscience, and has attracted some attention from the machine learning community in recent years (e.g. Bengio et al. 2015, Scellier & Bengio 2016, Sacramento et al. 2018). It is therefore a suitable topic for ICLR. Of course the good performance of single-layer random projections is not surprising, because it is essentially the idea of the Extreme Learning Machine, and this concept has been well studied also for neuromorphic approaches (e.g. Yao & Basu, 2017), and versions with local receptive fields exist as well (Huang et al. 2015 "Local Receptive Fields Based Extreme Learning Machine"). While the comparison of different unsupervised methods on MNIST is somehow interesting, it fails to show any deeper insights because MNIST is a particularly simple task, and already the CIFAR 10 results are far away from the state-of-the-art (which is >96% using CNNs). Another interesting comparison that is missing is with clustering weights, which has shown good performance for CNNs e.g. in (Coates & Ng, 2012) or (Dundar et al. 2015), and is also unsupervised.

The motivation is not 100% clear because the first experiment uses spikes, and shows a non-negligible difference to rate models (the authors claim it's almost the same, but for MNIST differences of 0.5% are significant). All later results are purely about rate models. The authors apparently did not explore e.g. conversion techniques as in (Diehl et al. 2015) to make the spiking results match the rate versions better e.g. by weight normalization.

I would rate the significance to the SNN community as average, and to the entire ICLR community as low. The significance would be higher if it was shown that this method scales to deeper networks or at least can be utilized in deeper architectures. Scrutinizing the possibilitites with random projections on the other hand could lead to more interesting results. But the best results here are obtained with 5000 neurons with 10x10 receptive fields on images of size 28x28, thus the representation is more than overcomplete, and of higher complexity than a convolution layer with 3x3 kernels and many input maps.

Because the results provide only limited insights beyond MNIST I can therefore not support acceptance at ICLR.

Pros:
+ interesting comparison of unsupervised feature learning techniques
+ interesting topic of bio-plausible deep learning

Cons:
- only MNIST, no indications if method will scale
- results are not better than state-of-the-art


Minor comments:

The paper is generally well-written and structured, although some of the design choices could have been explained in more detail. Generally, it is not discussed if random connections have any advantage over other spiking models in terms of accuracy, efficiency or speed, besides the obvious fact that one does not have to train this layer.

The title is a bit confusing. While it's not wrong, I had to read it multiple times to understand what was meant.

The first sentence in the caption for Fig. 2 is also confusing, mixing the descriptions of panel A and B. Also, in A membrane potentials are shown, but the post-membrane potential seems to integrate a constant current instead of individual spikes. Is this already the rate approximation of Eq. 2? Or is it because of the statement in the caption that they both receive very high external inputs. In general, the figures in panel A and B do not make the dynamics of the network or the supervised STDP much clearer.

Principal Component Analysis and Sparse Coding are done algorithmically instead of using a sort of nonlinear Hebbian Learning as in Lillicrap 2016. It would have been interesting to see if this changes the comparatively bad results for PCA and SC.

In Fig. 3, the curve in the random projections case is not saturated, maybe it would have been interesting to go above n_h = 5000. As there are 784 input neurons, a convolutional neural network with 7 filter banks already would have around 5000 neurons, but in this case each filter would be convolved over the whole image, while with random projections the filter only exists locally.

In Eq. 1, the notation is a bit ambigous: The first delta-function seems to be the Dirac-delta for continuous t, while the second delta is a Kronecker-delta with discrete t.

In A.1 and A.4.2 it is stated that the output of a layer is u_{t+1} = W u_t + b but I think in both cases it should be W a_t + b where a_t = phi(u_t). Otherwise, you just have a linear model and no activations.

In Table 3, a typo: "eq. equation"

---

> ### Author Response · Authors · 2018-11-22
> **Reply to AnonReviewer2**
>
> Thank you very much for the very detailed review.
>
> 1. We agree and are aware that our model is very similar to extreme learning architectures as already mentioned and cited in our paper. We missed the reference to patchy extreme learning and will include it; thanks. We also agree that MNIST is a particularly simple task and that already CIFAR10 shows the limits of “shallow” unsupervised learning and random projections. This is basically the main point we want to make (as literally written in the paper): MNIST is not suitable for evaluating deep learning capabilities since it can be solved by our simple system.
>
> 2. Thanks for pointing to the clustering weights papers. We will cite them in the related work section. However we could not find MNIST benchmarks for them and thus cannot compare them as in Table 8. We believe that the K-winner take all approach by Diehl 2015 (Table 8) is similar to such clustering methods.
>
> 3. Our motivation is twofold: 1) bio-plausibility, hence the purely spiking STDP model, and 2) systematic comparison of unsupervised methods (using rates to save computation time). All models share the important bio-plausible factors of local learning rules and online learning. We agree that 0.5% is significant on MNIST. This is why we explicitly did explore spike conversion techniques as in Diehl 2015 (see also methods section in the appendix). We found that plugging weights learned with a rate model leads to the same performance as pure rate models. We thus gained confidence that the remaining gap between rate and spike model is due to 1) spike artefacts in the STDP rule and transient phases and 2) much shorter simulation time (13 (spiking) vs 130 (rate) epochs) due to long simulation times. We work on optimising the spiking implementation.
>
> 4. Our aim was not to beat benchmarks but to challenge benchmarks with a particularly simple, bio-plausible system. Thus the limitation to a shallow architecture. We would like to point out again that this simple model outperforms many comparatively deep models. We do not doubt the usefulness of deep architectures but doubt the common practice of testing deep models on MNIST.
>
> 5. We did explore different patch sizes (spanning the whole range between single pixel and full image) and hidden layer sizes up to 10^5 hidden neurons (mentioned in the text; not plotted). We obtain that random projections reach backprop performance. This response also addresses the minor comment on hidden layer size later in the review.
>
> 6. On complexity: We have the same number of hidden neurons (5000) as one convolutional layer with 7 filter banks (not including pooling neurons or fully-connected layers). Weight sharing reduces the complexity of the architecture a lot, however, this cannot be considered as a bio-plausible model. Since our random projections are fixed and not learned, the number of trained parameters of our model is rather low, even compared to most CNNs.
>
> 7. Our contribution beyond MNIST (i.e. our comparison to CIFAR10) is indeed small but important: Even though machine learning moved to harder data sets, MNIST still seems to be important to test models for biological plausible deep learning. Our model shows that already switching to CIFAR10 could reveal real deep learning capabilities much more significantly.
>
> (MINOR COMMENTS)
>
> 8. On the LIF/STDP figure: Indeed, the post synaptic neuron is integrating up a high constant sub-threshold current. This bias current facilitates the learning. The growing feedforward input due to the growing weight can still be seen as small steps in the post-potential and in the post-synaptic spikes (otherwise the post-potential would stay below threshold).
>
> 9. PCA and Sparse Coding were already implemented with nonlinear Hebbian learning rules as suggested by the reviewer. Originally, we implemented PCA both as Hebbian learning and algorithmically. After checking for consistency the faster algorithmic way was chosen (as described in the paper; see also method section).

---

> > ### Comment · AnonReviewer2 · 2018-11-26
> > **Reply to authors's response**
> >
> > 1. " MNIST is not suitable for evaluating deep learning capabilities since it can be solved by our simple system."
> > Agreed, but I think this was already clear before your benchmark and I think the paper does not elaborate enough on this point if this is supposed to be an important finding of the paper.
> >
> > 3.
> > You talk about conversion techniques from rate-based to spiking and as you said, you cite Diehl 2015. But it is as far as I can see never stated that you use the proposed normalization scheme (data- or model-based normalization). But this is the key contribution of that paper: Not simply training an ANN and taking the same weights for the SNN,  but normalizing the weights beforehand.
> >
> > 4.
> > See answer 1.
> >
> > 5.
> > If you mention it in the text and explore it, you should also plot it, because in this case it would not have taken much space and not plotting things that could easily have been plotted is in my opinion not considered a good practice.
> >
> > 6.
> > Agreed that the number of trained parameters is low. Still, the number of neurons/synapses is not, and this is also important.
> >
> > 7.
> > Again, see answer 1.
> >
> > In conclusion, I still think the submission is interesting, but needs to be more organized: If the major result is that random projections can achieve good results, it has to be shown that this is not only true for MNIST. If the major result is that MNIST in general is not suitable as a benchmark for bio-plausible deep learning, more evidence in general is needed. Using both the rate- and spike model is also more confusing than enlightening and maybe it would be better to stick to only one. The claim that the rate model is an approximation of the spike model has to be solidified.

---

### Official Review · AnonReviewer1 · 2018-11-03
**No significant contribution**

**Rating:** 3
**Confidence:** 4

**Review:**

In this work authors benchmark a biologically plausible network architecture for image classification. The employed architecture consists of one hidden layer, where input to hidden layer weights W1 are either trained with PCA or sparse coding, or are kept fixed after random initialization. The output layer units are modeled as leaky integrate-and-fire (LIF) neurons and hidden to output connections W2 are tuned using a rate model that mimics STDP learning dynamics in the LIF neurons. The authors compare classification results on MNIST and CIFAR10 datasets, where they also include results of an equivalent feed-forward network that is trained with standard error backpropagation.

The authors find that in the bio-plausible network with a large hidden layer, unsupervised training of input to hidden layer weights does not lead to as good of a classification performance as achieved through fixed random projections. They furthermore find that localized patch-style connectivity from input to hidden layer further improves the classification performance.

Overall the paper is well-written and easy to follow, but I fail to see any significant contribution in this work. As compared to the findings of Hubel & Wiesel, how bio-plausible are random projections for low-level feature representation? One may also argue that unsupervised tuning of W1 may require a lot more training data than available in MNIST or CIFAR10. The authors also need to take the capacity of their network into account; they draw conclusions based on a biologically-plausible network, but one that only has two feed-forward layers. It is hard to imagine that a more accurate biologically-plausible vision model would prefer random projections over low-level feature extractors that are well-tuned to the input statistics.

Regarding the observation that localized fields perform better than densely connected layers, I find it simply in line with physiological findings (starting from the work of Hubel & Wiesel) and artificial neural network architectures they inspired like CNNs.

---

> ### Author Response · Authors · 2018-11-22
> **Reply to AnonReviewer1**
>
> Thank you for your review.
>
> 1. We simulate both pure spiking models using online STDP and pure rate models.  In addition we investigated a rate model that mimics LIF and STDP which offers a mapping from rate to spike models. In the review these concepts seemed to be confused. If the paper is unclear about this point, we would appreciate further feedback.
>
> 2. We agree that random connectivity is a very coarse model for low-level feature representation. And we see the main point/contribution of our paper exactly therein: random features are powerful enough to challenge the majority of the more elaborate, bio-plausible models we summarised in Table 8, including some that explicitly pick up ideas by Hubel & Wiesel or CNNs. We do not propose a model for the visual system but want to give a minimalistic benchmark and question the use of the MNIST data set for benchmarking.
>
> 3. Low-level feature extractors that are well-tuned to the input statistics as e.g. obtained with sparse coding or PCA do indeed lead to better performance than random projections for small numbers of hidden neurons (See Figure 2). We were also surprised that for higher hidden layer sizes random projections perform as least as good as the features obtained with sparse coding. We also experienced that sparse coding is rather sensitive to hyper-parameter tuning, which was an additional reason for us to stay away from trying to implement sparse coding in spiking networks. Your valuable questions prompted us to try an alternative: random Gabor patches, i.e.  instead of sampling each weight independently from a Normal distribution, we sample uniformly in a certain interval the angles, lengths and frequencies etc. of Gabor functions to define our hidden layer weights. Preliminary experiments with such random Gabor projections lead to even better performance than the pure random projections for all hidden layer sizes.  But, as we mentioned in point 2, our original goal was to show that even a very simple model performs comparable to (much) more elaborate bio-plausible models.
>
> 4. It is true that the early work by Hubel & Wiesel has already given evidence for localised connectivity in the visual system and we agree that our model is inline with these findings. However, it is not a priori clear that this leads to improved performance in supervised object recognition tasks, especially since the simple & complex cell model by Hubel & Wiesel is entirely task agnostic. CNNs use not only local connectivity, but also weight sharing, i.e. a reduction of trainable parameters, that is not present in our random filters.

---

### Official Review · AnonReviewer3 · 2018-11-07
**Well executed, but not exploring challenging questions**

**Rating:** 5
**Confidence:** 3

**Review:**

Summary:
The authors propose a benchmark of biologically plausible ANNs on the MNIST dataset with an emphasis on local learning rules (ruling out backpropagation, and enforcing small receptive fields). They find that random projection (RP) networks provide good performance close to backpropagation and outperform other local learning rules based on unsupervised learning.



Evaluation:
A well-executed work but with major limitations: it is based mostly on MNIST, analysis of spiking network is limited, and deep biologically plausible learning rules are not investigated.

Detailed comments:

While the paper reads well, choosing how to evaluate the contribution for such benchmark paper is a bit difficult, as the novelty is by definition more limited than in papers proposing a new approach.
In the following I chose to focus on what information such benchmark may bring to the field for addressing to challenges ahead.

1.	Strengths
The authors made the effort of implementing several biologically plausible learning rules, including Feedback alignment, and sparse coding. In particular, the idea of using local unsupervised learning rules as baselines for learning the hidden layer is a good idea to extend the range of tested algorithms.

2.	“Easy” dataset
It is unclear to me in which way MNIST result can help evaluate the next challenges in the field. While it is good to know that simple algorithms can achieve close to state of the art, I am not sure this is enough for a paper submitted in 2018. Ideally, most of the analysis could be reproduced at least for CIFAR10 (as the authors started to do in table 2).

3.	Limited architectures
Most of the analysis is restricted to one single layer. However, biologically plausible algorithms have also been proposed that can in principle apply to multiple layers. In addition to feedback alignment (implemented in the manuscript in the single hidden layer case), you can find relatively simple approaches in the literature, for example
“Balduzzi, David, Hastagiri Vanchinathan, and Joachim M. Buhmann. Kickback Cuts Backprop's Red-Tape: Biologically Plausible Credit Assignment in Neural Networks. AAAI. 2015.” Given the dominant view that depth is key for learning challenging datasets, not exploring this option at all in a benchmark seems a significant weakness.

4.	Spiking networks
While the authors seem to emphasize spiking as an important aspect of biological plausibility (by using LIF neurons and STDP). The challenges of such approaches seem to be largely unaddressed and the main take home message is a performance similar to the corresponding rate models. It would be very interesting, for example, to see how many spikes (or spikes per neurons) are need per example to achieve a robust classification.

5.	Overall objective behind biological plausibility
Extending the previous point, the results are to some extent limited to accuracy. If one wishes to achieve biological plausibility, more aspect can be taken into consideration. For example:
-	During test: the average number of activated neurons, the average number of activated synapses.
-	During training: the overall number of activations needed to train the algorithm.
In relation to these consideration, a more concrete discussion about the potential benefits of biological plausibility would be helpful.

---

> ### Author Response · Authors · 2018-11-22
> **Reply to AnonReviewer3**
>
> Thank you for this careful review.
>
> 1. We agree that MNIST is not sufficient to evaluate upcoming challenges in the field, e.g. bio-plausible deep learning. This is exactly what we want to point out with our simple model. We agree that MNIST is somewhat outdated in 2018.  However, we would like to note that MNIST is still used extensively in the bio-plausible field (see Table 8), hence our work could promote the important point that MNIST is too easy, just as the reviewer pointed out. We applied the best performing methods to CIFAR10 and found major limitations of our simple model and the  models based on unsupervised learning with a single hidden layer.  We suggest the use of CIFAR10 in the future. We did not run the spiking model on CIFAR10 due to limited computational resources and because we are confident to reach similar accuracies as with the rate model, as we saw for MNIST. If you expected additional analysis for CIFAR10, we would be happy to receive further feedback.
>
> 2. The restriction to a simple model with one layer is on purpose as mentioned above; we do not attempt to beat benchmarks but challenge existing models with a simpler model. We do not claim that our model is a model for deep learning beyond one hidden layer. We are aware of the literature proposing multi-layer bio-plausible deep learning and tried to put them into context in our paper (see section related work and Table 8). Most of the algorithms do not perform better than a single layer architecture. Also, most of these models do not scale to deep architectures (Bartunov 2018, cited). Thank you for the reference Balduzzi 2015.
>
> 3. We agree that assessing bio-plausibility could be done in more ways than only looking at accuracy. Your suggestions to estimate spike numbers per (output-) unit for robust classification are very interesting, also regarding energy consumption in neuromorphic implementations; thanks.

---

### Meta-Review · Area_Chair1 · 2018-12-13
**Interesting demonstration of a biologically plausible neural network but analysis and compelling experiments are lacking**

**Confidence:** 5
**Recommendation:** Reject

**Metareview:**

This paper presents a biologically plausible architecture and learning algorithm for deep neural networks.  The authors then go on to show that the proposed approach achieves competitive results on the MNIST dataset.  In general, the reviewers found that the paper was well written and the motivation compelling.  However, they were not convinced by the experiments, analysis or comparison to existing literature.  In particular, they did not find MNIST to be a particularly interesting problem and had questions about the novelty of this approach over past literature.  Perhaps the paper would be more impactful and convincing if the authors demonstrated competitive performance on a more challenging problem (e.g. machine translation, speech recognition or imagenet) using a biologically plausible approach.